# Diagnostic Interval in Extranodal Non-Hodgkin Head and Neck Lymphomas

**DOI:** 10.3390/jcm11030853

**Published:** 2022-02-06

**Authors:** Alba Bello Castro, Juan Seoane, Máximo Francisco Fraga Rodríguez, Francisco Gude Sampedro, Javier Seoane Romero, Benjamín Martin-Biedma, Pablo Castelo-Baz

**Affiliations:** 1Department of Surgery and Medical-Surgical Specialties, School of Medicine and Dentistry, Universidade de Santiago de Compostela, Santiago de Compostela, 15705 La Coruña, Spain; juanmanuel.seoane@usc.es (J.S.); javier.seoane.romero@gmail.com (J.S.R.); b.martinbiedma@gmail.com (B.M.-B.); pablocastelobaz@hotmail.com (P.C.-B.); 2Pathological Anatomy Unit, Universidade de Santiago de Compostela, 15705 La Coruña, Spain; maximo.fraga@usc.es; 3Unit of Clinical Epidemiology, Clinical University Hospital of Santiago de Compostela, 15705 La Coruña, Spain; francisco.gude.sampedro@sergas.es

**Keywords:** non-Hodgkin lymphomas, head and neck, diagnostic interval, diagnostic delay, dysphagia

## Abstract

The aim of this study is to measure the diagnostic interval (DI) of primary extranodal non-Hodgkin lymphomas (PE-NHL) affecting the head and neck and to discover any associated factors. With this aim, we performed a retrospective observational study in northwestern Spain on patients diagnosed between 1 January 2005 and 1 January 2016. A search was made across the electronic health records of the public health system of this region (SERGAS). DI was used as the dependent variable, and different clinicopathological data of the corresponding patients and tumors were analyzed as exposure variables. PE-NHLs were mostly located in Waldeyer’s ring, and they presented a B phenotype and had a median DI of 65 days. Shorter diagnostic intervals were observed in (1) PE-NHL patients who had comorbidities (*p* = 0.02), (2) PE-NHL that caused symptoms of dysphagia (*p* = 0.04), (3) tumors with the highest proliferative activity (Ki67 > 80%) (*p* = 0.04), and (4) tumors diagnosed in the advanced stages of the disease (*p* = 0.004). Univariate analysis revealed a significant association between dysphagia and a shorter DI. We conclude that raising awareness about these neoplasms and warning about the presenting symptoms can contribute to earlier diagnoses of these tumors and to better outcomes.

## 1. Introduction

Lymphomas are solid tumors of the immune system, and among these, non-Hodgkin lymphomas (NHLs) account for 90% of the total. NHLs are a broad and heterogeneous group of neoplasms of the lymph nodes and extranodal lymphatic system produced by clonal lymphocyte proliferations [1,2]. Unlike Hodgkin lymphomas, NHLs have an erratic dissemination pattern and very variable evolutions, ranging from highly proliferative and rapidly lethal forms to indolent subtypes that are compatible with a good quality of life without treatment [1,2]. NHLs have an increasing incidence in many regions (western Europe, India, Brazil, and Japan), reaching 6.7 cases/100,000 inhabitants/year in males and 4.7 cases in females in a global (worldwide) context [2,3].

A long interval until diagnosis and treatment in symptomatic cancers (breast, colorectal, head and neck, testicular, and melanoma) seems to be associated with poor outcomes [4]. In hematological cancers, diagnostic and therapeutic delays are associated with an increase in complications, and in particular, such delays are independent prognostic factors in aggressive forms [5].Despite the fact that early diagnosis of symptomatic cancers is a worldwide priority [6], the literature on time intervals in chronic hematological malignancies is very scarce [7], and, specifically, the magnitude and the determining factors of the diagnostic interval (“diagnostic delay”) in PE-NHL are unknown. Considering this lack of studies, our objectives were to measure the diagnostic interval in PE-NHL of the head and neck and to analyze the clinical and pathological variables associated with this time interval. 

## 2. Materials and Methods

A retrospective observational study was designed to identify PE-NHLs diagnosed in Galicia (northwestern Spain) between 1 January 2005 and 1 January 2016 by searching the IANUS system (electronic records of the public health system SERGAS platform), which includes information from hospitals included in the public network of the autonomous community. This system provides healthcare to its 2,700,000 inhabitants. We also performed an active search in the clinical departments that were involved.

Primary extranodal NHLs are defined as lymphomas that are localized in the extranodal compartment. In the case of lymph node involvement, this is minor and is associated with a clinically dominant extranodal component. The patients in our study strictly met these criteria [8,9,10]. After staging, those classified as having stages III or IV were only deemed primary extranodal if they had minor lymph nodes and/or bone marrow involvement.

For the analysis, the following independent variables were considered: age, sex, smoking, comorbidity, presenting symptoms (symptoms reported at the presentation by the patient who was later diagnosed of PE-NHL) [11] such as B-symptoms (temperature > 38°, weight loss > 10% in the 6 months preceding admission, and night sweats), the number of involved sites, histological subtypes of NHL [12,13], Ki-67 (as a cell proliferation factor ≥80% vs. <80%), stage (Ann Arbor staging system) [14], Eastern Cooperative Oncology Group (ECOG) performance status [15], and International Prognostic Index (IPI) for NHL [4].

The diagnostic interval (the date of the first symptom/sign until the date of histological diagnosis) was considered as a dependent variable in the model of pathways to treatment [16]. The investigation was carried out with the authorization of the Research Ethics Committee of the Xunta de Galicia (CEIm-G) with the code MFR-RIT-2018-01 (2018/596) and was in full accordance with the Declaration of Helsinki.

## 3. Statistical Analysis

Single-variable descriptive analysis was performed using frequencies for qualitative variables and using the mean (standard deviation), the median (interquartile range), and the 90th percentile for quantitative variables. The non-parametric Mann–Whitney U and Kruskal–Wallis tests were used in order to compare the differences in time to diagnosis according to tumor and patient baseline characteristics.

By using the median as a cutting point, we dichotomized the variable “time to diagnosis” as either a short time to diagnosis (<65 days) or a long time to diagnosis (>65 days). Thereafter, a univariate analysis was performed applying logistic regression using the dichotomized variable as a dependent variable. Moreover, a multivariate logistic regression was made using the diagnosis interval (DI) as a dependent variable.

## 4. Results

A total of 139 patients diagnosed with symptomatic PE-NHL met the inclusion criteria in the study with a male/female ratio of 1.01 and a mean age of 67.5 ± 16.8 years. The median diagnostic interval for the sample was 65 days (IQR, 32-119).

A total of 131 patients (94%) were in a good clinical condition (ECOG performance score: 0–1), and 57% of cases were categorized into a low-risk prognostic group (IPI score: 0–1), 36–50% of cases had an intermediate risk (IPI score: 2–3), and only 6% of cases were classified as having high risk (IPI score: 4–5). A total of 36 patients (26%), had associated comorbidity and 8 (6%) were HIV+ patients. Patient characteristics are summarized in Table 1.

PE-NHLs were preferentially located in Waldeyer’s ring (47.5%) and in salivary glands (24.5%). The most common presenting symptoms were swelling (52.5%) along with pain (20%). Extranodal masses were measured to have a diameter of 36 mm (34–101) at the time of diagnosis, although only six patients (4.3%) had a bulky mass (>70 mm in diameter). The vast majority (91.7%) had a B phenotype, and Diffuse B-Large Cell Lymphomas (DLBCL) accounted for 58% of the sample. The characteristics of PE-NHL are summarized in Table 2.

Some patient subgroups presented shorter diagnostic intervals (Table 1 and Table 2). These were: (1) PE-NHL patients who had comorbidities (*p* = 0.02), (2) those cases with dysphagia as presenting symptoms (*p* = 0.04), (3) tumors with the highest proliferative activity (Ki67 > 80%) (*p* = 0.04), and (4) those diagnosed at advanced stages of the disease (*p* = 0.004).

Univariate regression unveiled a significant association between DI and presenting symptoms, such as dysphagia (HR = 0.206, 95% CI 0.063–0.677), but not between DI and any of the following: the Ki-67 value, NHL stage, comorbidity, age, sex, tobacco consumption, associated pathology, presence of B-symptoms, raised LDH, ECOG performance status, IPI score, the number of involved sites, tumor size or histological subtypes (Table 3).

Additionally, we noted a significant association between DI and the presenting symptoms in the multivariate regression (Table 4).

## 5. Discussion

### Study Limitations and Strengths

As with any study, there are some weaknesses that must be taken into consideration for adequate interpretation of our data. These points are: (1) differences in geographical distribution of the different NHL subtypes, (2) a potential recall bias, and (3) potential misclassification.

For the first point, epidemiologic data indicate the existence of remarkable differences in the incidence of some NHL subtypes between geographical locations and race; therefore, the extrapolation of data from our cohort should be performed cautiously, particularly to regions with a non-Caucasic population [2].

Based on the retrospective nature of the study, a potential recall bias should be assumed, which could compromise the information recalled by the patients. However, this bias could also affect prospective diagnostic delay designs, as a random design is not possible for ethical reasons. In order to minimize this potential information bias, hospital data were checked against primary healthcare records, which are based on symptom information that is less prone to memory bias than self-reported information [11].

Furthermore, the diagnostic interval could be subject to differential misclassification; non-random recalling of dates when symptoms are vague occurs in indolent forms of PE-NHL. Our observational study included a large patient sample recruited with a high inclusion rate (94.4%), making the presence of selection biases unlikely. However, the fact that researchers involved in the design of the study had also undertaken data retrieval tasks may have resulted in an information bias, but the type of data used in our study and the retrospective nature of our investigation makes the existence of this particular systematic error highly unlikely.

Among the strengths of the analysis, it should be noted that a wide multicenter sample was analyzed, which is highly representative of the study population. Furthermore, restrictive criteria in the case definition (PE-NHL) were used, which avoids misclassifications associated with the use of liberal criteria of PE-NHL [17]. The definitions of time points and diagnostic intervals of the model of pathways to treatment (the Aarhus statement) as well as the requirements of the Aarhus checklist used to estimate the date of the first symptom and the date of the diagnosis [16] have also been followed. In addition, regression analysis has been carried out to control for confounding factors associated with observational studies.

A higher burden of comorbidities has been shown to behave as a diagnostic delay predictor in DLBCLs. However, potential DI inaccuracies must be assumed in the context of comorbid conditions [11,15].

Presenting symptoms of hematological malignancies are usually vague and nonspecific, with a broad symptom signature and diagnostic difficulty [11]. Specifically, lymphomas are considered to be of intermediate diagnostic difficulty, and patients usually undergo multiple primary care consultations prior to diagnosis, thereby lengthening the diagnostic time interval [11]. The few available studies, focused on NHLs at any location, are of European origin and found a median DI in a range of 51 to 132 days [5,7]. Meanwhile, the only head and neck NHL study, which considered extranodal lymphomas, showed a median DI of 90 days, which is far higher than that found for PE-NHLs in this series.

According to our data, lymphoma histological grade plays a key role in diagnostic delay, with aggressive tumors such as DLBCL being more prone to be diagnosed earlier than low-grade tumors, which can have an indolent course and might be easily confounded with benign lesions. Indeed, we observed a shorter DI associated with tumors with higher proliferative activity (Ki67 > 80%), which is consistent with similar reports of NHL affecting other anatomic locations and of PE-NHL affecting the head and neck [18]. Furthermore, the subtypes with greater clinical aggressiveness included in our series (e.g., DLBCL) presented a lower DI than mucosa-associated lymphoid tissue (MALT) lymphomas and Follicular Lymphomas, which usually follow an indolent course. This same distribution has also been observed for NHL from other locations [19].

Presenting symptoms in our series determine the diagnostic interval. Particularly, the following symptoms provided an earlier diagnosis: dysphagia and pain of the NHLs of Waldeyer’s ring, observable tumors in the oral cavity, and the presence of B-symptoms, whereas nonspecific symptoms generated by the sinonasal lymphomas provide the greatest intervals until diagnosis [5,7]. These findings are consistent with what happens in different types of lymphomas in different locations [5,7,15].

In concordance with our series, different neoplasms, such as breast, lung, colorectal, ovarian, and oral cancer, have shown a paradoxical association between shorter diagnostic intervals and having an advanced stage at diagnosis [20]. This counterintuitive association seems to be justified by indication confounders (waiting time paradox) where professionals prioritize severely ill patients for diagnosis and where the most proliferative tumors are the ones that have shown the fewest intervals until diagnosis.

The hematimetry findings were not relevant and do not appear to contribute to diagnostic suspicion, probably due to the extranodal nature of NHL in our series, which excluded patients with significant lymph node involvement and/or bone marrow infiltration. In contrast, hypogammaglobulinemia appeared to be more common in indolent NHL, and the elevation of markers of proliferation such as LDH and beta2-microglobulin [21] in biochemical analysis appeared in a quarter of cases. Thus, in the extranodal tumors in the head and neck, the elevation of LDH suggests the presence of pathology of lymphoid origin and a high degree of aggressiveness.

## 6. Implications for Clinicians and Researchers

The head and neck PE-NHLs have few specific symptoms and are dependent on the histological subtype and tumor location, which is the reason why they generate a low diagnostic suspicion and wide temporal intervals until definitive diagnosis (“diagnostic delays”). These circumstances and the heterogeneous clinical presentation of NHLs represent a diagnostic challenge for ENT specialists.

Signs and symptoms of PE-NHL may be similar to that of squamous cell carcinomas of the head and neck, and they are only histologically differentiable [17]. In the case of PE-NHL, the symptomatology is very dependent on the location and aggressiveness of the histological subtype [22]. In our series, frequent involvement of Waldeyer’s ring and the oral cavity by aggressive subtypes (clear predominance of DLBCL) conditioned the shortest diagnostic intervals. By contrast, PE-NHLs, which had more trivial symptoms and glandular seating, were histologically associated with a high proportion of low-grade B lymphomas (MALT and follicular lymphomas), representing the greatest “diagnostic delays”. Based on the above, in order to achieve an early diagnosis and better outcomes, it seems necessary to increase our diagnostic suspicion towards this neoplasm and to use well-established clinical protocols. In the case of NHLs, the use of algorithms of a recommended diagnostic workup has been suggested, including the careful endoscopic examination of upper cavities, a deep biopsy of extranodal diseases, blood analysis (LDH level, HIV serology, CD4+ cell count, etc.), and complementary examinations (orthopantomography, CT-Scan, MRI, etc.) [23,24]. Particularly, this awareness should be applied to risk groups associated with long diagnostic intervals.

Future research is also necessary to develop strategies based on the early diagnosis of symptomatic NHLs in the head and neck. For this purpose, multicenter studies should be designed which consider the ethnicity and geographic area of the sample that has a clear case definition and that follows the conceptual framework (model of pathways to treatment) and the standards of the Aarhus Declaration developed for improved design on early cancer diagnosis.

## 7. Conclusions

Head and neck PE-NHLs have a heterogeneous clinicopathological presentation with different presenting symptoms, although their B phenotypes and their locations in Waldeyer’s ring predominate. Dysphagia behaves similar to an “alarm symptom” and appears to be associated with earlier diagnoses (shorter DI). However, the interval to diagnosis shows a wide space for improvement. Raising awareness about these neoplasms and warning about any presenting symptoms can contribute to obtaining early diagnoses and better outcomes.

## Figures and Tables

**Table 1 jcm-11-00853-t001:** Comparison of diagnostic intervals according to patient characteristics.

Extranodal NHL		Diagnostic Interval			
Variable	*N* (%)	Mean (SD)	Median (Interquartile Range)	90%	*p*-Value
Age					
<60	40 (29%)	135 (29)	83 (33–172)	299	0.198
≥60	95 (68%)	95 (11)	61 (31–104)	226	
Gender					
Male	70 (50.4%)	95 (13)	62 (26–117)	255	0.308
Female	69 (49.6%)	118 (19)	68 (37–138)	244	
Tobacco					
Smoker	109 (78.4%)	104 (13)	62 (32–116)	238	0.843
Non-Smoker	18 (13.9%)	108 (24)	70 (28–151)	300	
Former smoker	12 (8.6%)	128 (43)	98 (32–160)	457	
Comorbidity					
No	103 (74%)	117 (15)	69 (32–135)	302	0.027 *
Yes	36 (26%)	76 (11)	57 (31–117)	181	
Neoplasm					
No	116 (83.5%)	113 (13)	69 (32–129)	283	0.054
Yes	23 (16.5%)	74 (15)	53 (22–103)	211	
B/C hepatitis					
No	128 (92%)	107 (12)	63 (31–117)	263	0.614
Yes	11 (8%)	97 (16)	115 (51–122)	187	
HIV					
No	131 (94%)	108 (12)	63 (32–120)	258	0.164
Yes	8 (6%)	78 (17)	91 (26–120)		
B symptoms					
No	112 (80.5%)	109(13)	69 (32–120)	242	0.648
Yes	27 (19.5%)	98(21)	48 (31–122)	283	
LDH					
Normal	93 (67%)	106 (15)	62 (31–118)	238	0.931
High	30 (22%)	109 (20)	56 (31–164)	275	
ECOG					
0–1	131 (94%)	107 (12)	63 (32–120)	243	0.998
(2–3) and (4–5)	8 (6%)	107 (32)	97 (28–178)		
IPI					
0	79 (57%)	118 (17)	71 (36–133)	297	0.547
1	50 (36%)	91 (15)	48 (31–107)	239	
2	8 (6%)	112 (30)	90 (41–178)		

LDH: lactate dehydrogenase; ECOG: Eastern Cooperative Oncology Group; IPI: International Prognostic Index; *: statistically significant.

**Table 2 jcm-11-00853-t002:** Comparison of diagnostic intervals according to tumor characteristics.

Extranodal Non-Hodgkin Lymphoma	Diagnostic Interval				
Variable	N (%)	Mean (SD)	Median (Intercuartil Range)	90th Centile	*p*-Value
Symptoms					
Swelling	73 (52.5%)	113 (14)	76 (33–127)	302	0.049 *
Pain	28 (20%)	102 (23)	43 (27–139)	293	
Dysphagia	20 (14.5%)	63 (12)	45 (31–74)	191	
Nasal irritation	9 (6.5%)	95 (18)	92 (54–134)		
Lymphadenopathy	5 (3.5%)	83 (42)	59 (20–158)		
Odynophagia	4 (3%)	302 (255)	59 (26–821)		
Sites					
Waldeyer’s ring (ref)	66 (47.5%)	100 (19)	59 (31–102)	250	0.383
Nasal cavity/maxilar	23 (16.5%)	99 (21)	62 (29–135)	238	
Oral cavity	16 (11.5%)	75 (20)	37 (23–111)	229	
Glands	34 (24.5%)	139 (23)	100 (57–173)	357	
Size					
<36 mm	68 (49%)	129 (20)	76 (34–172)	308	0.061
≥36 mm	66 (48%)	86 (12)	57 (28–101)	182	
Stage					
(I-II)	95 (68.3%)	124 (16)	72 (36–140)	315	0.004 *
(III-IV)	43 (30.9%)	69 (10)	47 (25–104)	171	
Histology					
DLBCL (ref)	80 (58%)	86 (10)	59 (31–103)	173	0.085
FL	16 (12.5%)	83 (21)	56 (31–106)	241	
MCL	19 (14%)	164 (56)	94 (34–219)	417	
NKL	9 (6,5%)	157 (61)	89 (26–267)		
MALT	11 (8%)	165 (42)	135 (51–277)	431	
Otros	4 (3%)	66 (23)	66 (25–109)		
Ki67					
<80	40 (29%)	140 (29)	94 (37–190)	259	0.048 *
≥80	50 (36%)	79 (13)	47 (21–105)	172	

DLBCL: Diffuse large B-cell lymphoma; FL: Follicular Lymphoma; MCL: Mantle cell lymphoma; NKL: Natural Killer Lymphoma; MALT: mucosa-associated lymphoid tissue; *: statistically significant.

**Table 3 jcm-11-00853-t003:** Univariate analysis of related factors with shorter diagnosis interval.

Model	Components	B	SE	Wald	*p*-Value	OR (C.I 95%)
Interval-age	≥60	−0.492	0.381	1.67	0.196	0.611 (0.290–1.288)
Interval-gender	Female	0.433	0.341	1.611	0.204	1.542 (0.709–3.010)
Interval-tabaco	Non-smoker	−0.581	0.520	1.246	0.264	0.560 (0.202–1.551)
	Former smoker	−0.822	0.642	1.640	0.200	0.440 (0.125–1.547)
Interval-comorbidity	Comorbidity	0.019	0.387	0.003	0.960	1.020 (0.477–2.178)
Interval-neoplasm	Neoplasm	−0.511	0.466	1.202	0.273	0.600 (0.241–1.495)
Interval-B/C hepatitis	B/C Hepatitis	0.622	0.651	0.913	0.339	1.863 (0.520–6.676)
Interval-HIV	HIV	1.175	0.835	1.980	0.159	3.238 (0.630–16.636)
Interval- B symptoms	B symptoms	−0.872	0.450	3.754	0.053	0.418 (0.173–1.010)
Interval-LDH	LDH ≥ 418 u/L	0.247	0.423	0.341	0.559	1.280 (0.559–2.931)
Interval-ECOG	ECOG ≥ 2	0.015	0.728	0.000	0.983	1.015 (0.244–4.233)
Interval-IPI	IPI 1	0.500	0.365	1.881	0.170	1.649 (0.807–3.372)
	IPI 2	0.178	0.742	0.057	0.811	1.194 (0.279–5.117)
Interval-symptoms	Swelling	0.192	0.235	0.669	0.413	1.212 (-)
	Pain	0.095	0.448	0.45	0.832	1.1 (0.457–2.649)
	Dysphagia	−1.579	0.606	6.776	0.009 *	0.206 (0.063–0.677)
	Nasal irritation	−0.416	0.711	0.342	0.559	0.660 (0.164–2.658)
	Lymphadenopathy	−0.598	0.943	0.402	0.526	0.550 (0.087–3.490)
	Odynophagia	0.906	1.178	0.591	0.442	2.475 (0.246–24.925)
Interval-location	Nasal cavity	0.685	0.494	1.925	0.165	1.985 (0.754–5.226)
	Oral cavity	0.244	0.558	0.191	0.662	1.276 (0.427–3.810)
	Glands	0.361	0.424	0.727	0.394	1.435 (0.626–3.293)
Interval-size	≥36 mm	0.359	0.347	1.072	0.300	1.432 (0.726–2.827)
Interval-stage	III-IV	0.476	0.371	1.644	0.200	1.619 (0.778–3.333)
Interval-histology	DLBCL	0.150	0.224	0.449	0.503	1.162 (-)
	MCL	−0.402	0.552	0.530	0.467	0.669 (0.227–1.973)
	FL	−0.469	0.516	0.825	0.364	0.626 (0.228–1.720)
	NKL	−0.373	0.707	0.279	0.598	0.688 (0.172–2.753)
	MALT	0.032	0.646	0.002	0.960	1.033 (0.291–3.661)
	Other	−1.249	1.176	1.127	0.288	0.287 (0.029–2.876)
Ki67	≥80	−0.100	0.425	0.056	0.814	0.905 (0.394–2.079)

LDH: lactate dehydrogenase; ECOG: Eastern Cooperative Oncology Group; IPI: International Prognostic Index; DLBCL: Diffuse large B-cell lymphoma; FL: Follicular Lymphoma; MCL: Mantle cell lymphoma; NKL: Natural Killer/T-cell Lymphoma; MALT: mucosa-associated lymphoid tissue; *: statistically significant.

**Table 4 jcm-11-00853-t004:** Multivariate analysis of related factors with shorter diagnosis interval.

Model	B	Standard Error	Wald	*p*-Value	OR (C.I 95%)
Reference	−0.818	0.625	1.715	0.190	0.441 (-)
Stage	0.259	0.490	0.279	0.598	1.295 (0.495–3.388)
Symptoms	1.134	0.467	5.903	0.015 *	3.107 (1.245–7.754)
Localization	0.349	0.461	0.517	0.450	1.417 (0.574–3.502)
Morbidity	−0.384	0.514	0.557	0.455	0.681 (0.248–1.867)
Ki67	−0.035	0.456	0.006	0.938	0.965 (0.395–2.358)

*: statistically significant.

## Data Availability

Data can be shared upon request.

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
