# Peer review of "Diagnostic Interval in Extranodal Non-Hodgkin Head and Neck Lymphomas"

_jcm, 2022, doi:10.3390/jcm11030853_

Round 1

Reviewer 1 Report

Dear Authors,

The article: 'Diagnostic interval in extranodal non-Hodgkin head and neck lymphomas' was to measure the diagnostic interval in PE-NHL of head and neck, and to analyze the clinical and pathological variables associated with this time interval.

Add authors affiliations.

Abstract should be unstructurized.

References should be citted using [1,2]. Article should be prepared using MDPI guidelines.

English language and style are fine.

Punctuation mistakes should be corrected. 

p value is written in italics.

Add table with abbreviations.

References should be prepared in accordance with the MDPI guidelines

To sum up, article should be reconsider after major revision.

Author Response

Q1. Add author affiliations

A1. Author affiliations have been included

Q2. Abstract should be unstructured

A2. We have modified the abstract according to your recommendations

Q3. References should be cited dusing [1,2]. Article should be prepared using MDPI guidelines

A3. We have modified the cities of each reference according to your recommendations.

Q4. Punctuation mistakes should be corrected.

A4. We have corrected punctuation mistakes.

Q5. p-value is written in italics

A5. Sorry for that mistake, we have corrected it.

Q6. Add table with abbreviations

A6. A new table with abbreviations has been created.

Q7. References should be prepared in accordance with the MDPI guidelines.

A7. We have modified references according to MDPI guidelines.

Reviewer 2 Report

The Authors designed a retrospective, observational study on primary extra nodal non Hodgkin lymphomas of the head and neck area, with the aim of evaluating the diagnostic interval in this rather wide category of neoplasms.

Despite being an interesting topic for readers, some points need to be revised:

  1. multiple grammar errors need to be corrected in the text and Tables
  2. It should be clearly stated that one of the main factors influencing the different diagnostic interval is the histological type of non Hodgkin lymphoma; high grade lymphomas such as DLBCL usually have a completely different clinical presentation and behavior compared to indolent lymphomas for instance MALT lymphoma. Being a wide category of lymphomas including both low grade and high grade diseases, this should be clarified to the readers in order to understand the reason of the differences in diagnostic interval.
  3. In the discussion, the Authors should include all the main findings of the study in order to make clear to the reader which factors influence the diagnostic interval.
  4. The point of weakness of the study are not well clarified 

Author Response

Dear reviewer, thanks for your valuable comments. Find attached a point-by-point response letter to your questions and recommendations.

Q1. Multiple grammar errors need to be corrected in the text and Tables

A1. We have checked the paper and corrected such errors.

Q2. It should be clearly stated that one of the main factors influencing the different diagnostic interval is the histologic type of non Hodgkin lymphoma; high grade lymphomas such as DLBCL usually have a completely different clinical presentation and behavior compared to indolent lymphomas for instance MALT lymphoma. Being a wide category of lymphomas including both low grade and high grade diseases, this should be clarified to the readers in order to understand the reason for the differences in diagnostic interval. 

A2. We agree with Reviewer’s appreciation. We have modified the 5th paragraph of the discussion in order to better accommodate this issue. We have stated the importance of the different histologies in the interval to diagnosis as the main driver of diagnostic delay.

Q3. In the discussion, the Authors should include all the main findings of the study in order to make clear to the reader which factors influence the diagnostic interval.

A3. We agree with this recommendation and we have updated with review with all the findings described in the manuscript.

Q4. The points of weakness of the study are not well clarified.

A4.  We have extended our discussion about weaknesses so as to provide a more clear explanation of these. 

Round 2

Reviewer 1 Report

Dear Authors,

The authors did not follow the reviewer's guidelines. They wrote back that they had made corrections. There is no amendment in the text:

Add authors affiliations.

p value is written in italics.

Add table with abbreviations.

References should be prepared in accordance with the MDPI guidelines

To sum up, article should be reconsider after major revision.

Author Response

Dear Reviewer,

Thanks for your kind appreciations. Please, find attached a point-by-point response to your comments.

With best regards,

Alba Bello

Q1. Add authors affiliations.

A1. Author affiliations have been added.

Q2. p value is written in italics.

A2. We have checked the manuscript and found no p-value in italics, so we suspect that some difference between text editors could be behind this airtfact. Nevertheless, we would be happy to check this issue with the post-processing team once the paper is accepted.

Q3. Add table with abbreviations.

A3. A table with abbreviations has been added.

Q4. References should be prepared in accordance with the MDPI guidelines

A4. We have revised the reference style according to MDPI guidelines
